# Challenges in the Diagnosis of Parathyroid Cancer: Unraveling the Diagnostic Maze

**DOI:** 10.3390/reports6030040

**Published:** 2023-08-24

**Authors:** Mihaela Stanciu, Remus Calin Cipaian, Ruxandra Ristea, Corina Maria Vasile, Mihaela Popescu, Florina Ligia Popa

**Affiliations:** 1Department of Endocrinology, County Clinical Emergency Hospital of Sibiu, 550245 Sibiu, Romania; mihaela.stanciu@yahoo.com (M.S.); ruxandraszebin@yahoo.com (R.R.); 2Department of Endocrinology, Faculty of Medicine, Lucian Blaga University of Sibiu, 550169 Sibiu, Romania; 3Department of Internal Medicine, Faculty of Medicine, Lucian Blaga University of Sibiu, 550169 Sibiu, Romania; 4Department of Pediatric and Adult Congenital Cardiology, Bordeaux University Hospital, 33600 Pessac, France; 5Department of Endocrinology, University of Medicine and Pharmacy of Craiova, 200349 Craiova, Romania; mihaela.n.popescu99@gmail.com; 6Department of Physical Medicine and Rehabilitation, Faculty of Medicine, Lucian Blaga University of Sibiu, 550169 Sibiu, Romania; florina.popa@yahoo.com; 7Department of Physical Medicine and Rehabilitation, County Clinical Emergency Hospital of Sibiu, 550245 Sibiu, Romania

**Keywords:** parathyroid carcinoma, parathyroid hormone, early diagnosis

## Abstract

Parathyroid carcinoma (PC) is a rare and aggressive cancer affecting the parathyroid glands, presenting diagnostic and therapeutic challenges due to its rarity and overlapping features with benign parathyroid disease. This report describes the case of a 51-year-old patient with significantly elevated serum calcium levels, leading to further investigation. Imaging studies revealed a large nodular mass in the right inferior parathyroid gland. After undergoing right inferior parathyroidectomy, pathology confirmed parathyroid carcinoma. However, the patient experienced a relapse, detected via a neck MRI. This case highlights the importance of specific clinical features, such as rapid calcium increase, elevated parathyroid hormone (PTH) levels, and a large nodular mass, in suspecting malignancy. Differential diagnosis between carcinoma and adenoma can be challenging, and immunohistochemistry aids in diagnosis. Regular follow-up with calcium and PTH monitoring is essential for detecting recurrence. This case underscores the aggressive nature of parathyroid carcinoma and the importance of early diagnosis, surgical intervention, and thorough follow-up care for improved outcomes.

## 1. Introduction

Parathyroid carcinoma (PC) is a rare and aggressive type of cancer that affects the parathyroid glands. It accounts for less than 1% of all cases of primary hyperparathyroidism. It can be difficult to diagnose and treat due to its rarity and overlapping clinical and imaging features with benign parathyroid disease [1]. Unlike benign parathyroid disease, which is more common in women, parathyroid cancer occurs equally in both sexes and tends to be diagnosed at a younger age [2]. 

Patients diagnosed with PC have a poor prognosis, with 5- and 10-year overall survival rates of approximately 85% and 49%, respectively. This is primarily attributed to the frequently unmanageable, severe, and drug-refractory hypercalcemia accompanying the disease. Prognostic factors crucially influencing outcomes include tumor stage, invasion of adjacent tissues, and the presence of local and distant metastases at the time of initial diagnosis [3,4,5].

Parathyroid carcinoma does not show a gender bias. It affects both men and women equally, unlike benign parathyroid tumors, which are more commonly observed in women (with a male-to-female ratio of 1:4). This malignancy tends to present in individuals during their fourth to fifth decade of life, with a diagnosis typically occurring approximately 10 years earlier than parathyroid adenomas [6,7,8]. 

Recent research has identified a few clinical, biological, and imaging parameters that may suggest the presence of malignancy in patients with primary hyperparathyroidism. These patients typically present with more severe symptoms and higher calcium and parathyroid hormone levels than those with benign parathyroid adenomas [9].

Parathyroid ultrasound plays a valuable role in distinguishing malignant tumors from parathyroid adenomas. In the case of parathyroid cancer, ultrasound imaging typically reveals intralesional calcifications and a taller-than-wide shape, particularly in conjunction with markedly elevated serum parathyroid hormone levels (>1000 pg/mL) [10]. The primary treatment for parathyroid carcinoma is surgical resection, even in cases of metastatic disease. However, in some instances, the diagnosis of parathyroid carcinoma is only made after surgery [11].

In this case report, we present the challenging case of a 51-year-old patient who was diagnosed with parathyroid carcinoma after undergoing right inferior parathyroidectomy for primary hyperparathyroidism and experienced a relapse of the disease two months later, emphasizing the diagnostic difficulties encountered and the importance of recognizing specific clinical features that may raise suspicion for malignancy. We discuss the diagnostic workup, including laboratory investigations, imaging studies, and histopathological examination, highlighting each modality’s limitations and potential pitfalls. By elucidating the diagnostic challenges in parathyroid carcinoma, this case report aims to contribute to the existing literature and raise awareness among clinicians, aiding in the early detection and appropriate management of this rare malignancy.

## 2. Detailed Case Description

### 2.1. Patient Presentation

We present the case of a 51-year-old male patient referred to our clinic with no specific symptoms related to hypercalcemia or family or personal history of endocrine disease. Physical examination revealed a palpable nodule in the right thyroid lobe region, quite delimited and firm, of approximately 3 cm, high blood pressure (150/100 mmHg), and a body mass index of 33.5 kg/m^2^. The patient was addressed to our clinic due to identifying an elevated calcemia level (as serum) of 12.78 mg/dL during routine blood investigations. The tests also showed high values at the next two repeat tests within one month.

### 2.2. Clinical Investigations

Laboratory investigations revealed significantly elevated serum calcium levels of 12.78 mg/dL (normal range: 8.5–10.5 mg/dL) and increased parathyroid hormone (PTH) levels of 517 pg/mL (normal range: 10–65 pg/mL), serum phosphorus level of 1.9 mg/dL (2.5–4.5 mg/dL), serum alkaline phosphatase of 488 U/L(normal range 46–122 U/L), and 25-hydroxyvitamin D of 23 ng/mL (normal range < 30 U/L). These findings raised suspicion of primary hyperparathyroidism, leading to further diagnostic workup.

### 2.3. Imaging Findings

Ultrasonography examination of the cervical area revealed an enlarged, irregular, low-echo, and non-vascularized right inferior parathyroid gland. A distinct mass was detected in the right cervical region, displaying well-defined borders, heterogeneous echogenicity, predominantly low echoes, microcalcifications and fibrotic areas, and increased vascularity on Doppler imaging. The dimensions of the mass were measured at 3.7 cm × 5.0 cm × 2.8 cm, and it was not possible to definitively exclude the possibility of a parathyroid adenoma. Subsequently, a neck CT scan confirmed the presence of a nodular mass within the right inferior parathyroid gland (Figure 1).

### 2.4. Surgical Intervention

Based on the clinical presentation, laboratory results, and imaging findings, a diagnosis of primary hyperparathyroidism was suspected. The patient underwent a right inferior parathyroidectomy to remove the suspicious lesion. In the preoperative phase, the patient underwent saline solution, cholecalciferol, Furosemide, and Alendronic Acid treatment. Intraoperatively, the right inferior parathyroid gland was removed and was observed adherent to adjacent structures and exhibited firm consistency.

Before the procedure, the patient exhibited elevated parathyroid hormone (PTH) levels at 960 pg/mL and calcium at 14.1 mg/dL. After surgery, PTH and calcium levels decreased significantly to 4 pg/mL and 10 mg/dL, respectively. A more detailed timeline is presented in Figure 2.

### 2.5. Histopathological Examination

Macroscopic examination of the surgical specimen revealed a gray-colored, well-circumscribed nodular mass measuring 5 cm × 2 cm. Microscopic examination demonstrated a highly cellular lesion composed of tightly packed cells with round nuclei, forming nests within a fibrous stroma. Mitotic activity was noted at a rate of 2 mitotic figures per 10 high-power fields (HPF). No evidence of vascular invasion was observed. Immunohistochemical analysis revealed positivity for cyclin D1 and BCL-2, with a Ki-67 labeling index of 5%. Notably, the tumor showed negativity for TTF-1, thyroglobulin, and galectin 3. These findings were consistent with the diagnosis of parathyroid carcinoma. 

### 2.6. Disease Relapse

Despite the initial successful surgical intervention, the patient experienced a relapse of the disease after two months. Laboratory tests, including serum calcium, phosphate, and PTH measurements, indicated a relapse of the patient’s condition. Neck magnetic resonance imaging (MRI) was performed, which revealed the presence of a new nodular mass adjacent to the left thyroid lobe, located at the base of the cricoid cartilage. The mass measured 1.1 cm and exhibited imaging characteristics indicative of parathyroid carcinoma, including T2 isosignal, gadolinium enhancement, and hypersignal on diffusion-weighted imaging (DWI) without a corresponding signal on apparent diffusion coefficient (ADC) imaging. These imaging findings confirmed a possible relapse of the patient’s condition (Figure 3).

Considering the pathology results, which indicated parathyroid carcinoma, the recommendation for the patient was external radiation therapy instead of another surgical intervention. Subsequent SPECT/CT combined with 99mTc-sestamibi scanning imaging revealed no abnormal parathyroid lesions, including ectopic locations (Figure 4). A comprehensive follow-up care plan has been established to closely monitor the patient and mitigate the risk of any potential complications. 

### 2.7. Follow-Up

The patient, currently in a stable condition, continues to evolve favorably, with no disease relapse. Regular monitoring of calcium and parathyroid hormone (PTH) levels every two months has been essential to follow the ongoing health status. Notably, consultations were conducted at 6 months and 1 year, providing a comprehensive assessment of their well-being and affirming their sustained improvement. Figure 5 shows the values of calcium and PTH after 6 months and 1 year after the surgery.

## 3. Discussion

This case report highlights the diagnostic difficulties encountered in parathyroid carcinoma, even with various diagnostic modalities, including clinical evaluation, laboratory investigations, imaging studies, and histopathological examination. Despite the initial successful surgical resection, disease relapse occurred, emphasizing the aggressive nature of parathyroid carcinoma and the challenges in its long-term management.

Primary hyperparathyroidism is a condition characterized by excessive secretion of parathyroid hormone (PTH) due to an abnormality in the parathyroid glands. This leads to an elevated concentration of PTH in the blood, which stimulates increased renal calcium absorption and calcitriol synthesis. The actions of PTH also result in decreased phosphaturia and increased bone resorption. The cumulative effect of these actions is the hallmark biochemical phenotype of hypercalcemia and hypophosphatemia, decreased cortical bone density, increased urinary calcium levels, and the various clinical manifestations of chronic hypercalcemia. In most cases (75–80%), primary hyperparathyroidism is caused by one or more adenomas in normal parathyroid glands. In a minority of patients (20%), diffuse hyperplasia of all parathyroid glands may be present. Rarely, parathyroid carcinoma may be the underlying cause of primary hyperparathyroidism, affecting less than 1% of patients.

Parathyroid carcinoma is a rare malignancy that usually targets one of the four parathyroid glands, and its underlying causes are not thoroughly comprehended. The exact number of cases of parathyroid carcinoma reported in the literature is unclear and varies depending on the source. It is important to note that the rarity of this tumor makes it difficult to establish accurate statistics. This case involved a parathyroid carcinoma in the lower right parathyroid gland, and treatment included surgical removal of the affected gland. As the patient experienced a relapse early on, radiation therapy was used as a treatment option.

In contrast to benign parathyroid disease, which is more common in women than men (with a ratio of 3–4:1), parathyroid cancer affects both sexes equally. Furthermore, individuals with parathyroid carcinoma are usually diagnosed at an earlier age, approximately a decade earlier, than those with the benign variant of the disease. Specifically, the age of diagnosis for parathyroid carcinoma is commonly in the mid-40s, whereas for the benign forms, it typically occurs in the mid-50s [12]. This information highlights some key differences in the epidemiology of benign and malignant parathyroid conditions.

The etiology has been linked to mutations in HRPT2/CDC73 gene. Research by Howell et al. and Shattuck et al. in 2003 supports the role of HRPT2 in developing sporadic parathyroid carcinoma, as mutations in this gene were found in a significant proportion of patients with this type of tumor [13]. Cetani et al. also identified HRPT2 mutations in most parathyroid carcinomas but none in sporadic atypical adenomas. These findings underscore the importance of HRPT2 in the development of parathyroid carcinoma [14].

Distinguishing between parathyroid carcinoma and its more prevalent benign counterpart is crucial due to the significant morbidity and mortality associated with the diagnosis. Primary hyperparathyroidism, characterized by excessive parathyroid hormone (PTH) production, manifests symptoms resulting from the hormone’s effects on various body systems. Fractures, chronic bone pain, and spinal deformities like kyphosis may occur, reflecting skeletal involvement. Renal manifestations may include renal colic or renal failure. Gastrointestinal symptoms such as weight loss, loss of appetite, nausea, and vomiting may also manifest alongside polyuria and polydipsia. Ocular findings like conjunctival calcifications, band keratopathy, and hypertension can also be observed in approximately 50% of cases. In patients with parathyroid carcinoma, serum calcium levels are typically higher than those with benign primary hyperparathyroidism, and PTH levels are elevated by 3–4 times. While a palpable neck mass is uncommon in primary hyperparathyroidism, it is present in 30–76% of carcinoma cases [15].

Diagnosis of suspected parathyroid carcinoma requires the use of multiple imaging tests to locate the tumor accurately. The commonly utilized preoperative localization methods are 99mTc-sestamibi scanning, ultrasonography, CT scan, and MRI. Ultrasound is important to make the differential diagnosis with possible thyroid nodules positioned at the posteroinferior pole of the thyroid lobes [16,17].

Ultrasound of the cervical region of the neck can make it difficult to distinguish a parathyroid tumor from a thyroid nodule located on the posterior wall of the thyroid gland; the evaluation of the risk of thyroid and parathyroid malignancy must include the description of echogenicity, composition, shape, margins, and the presence of calcifications and associated lymphadenopathy [18].

When the ultrasound aspect is unclear, investigations with CT and SPECT/CT combined with 99mTc-sestamibi scanning are extended. Parathyroid carcinomas typically appear on ultrasounds as a solid mass with poorly defined borders. They tend to be larger than benign parathyroid adenomas and are usually greater than 2 cm in size. They are typically hypoechoic and may also contain calcifications. Additionally, parathyroid carcinomas are often highly vascular. Parathyroid carcinoma may invade surrounding structures, such as the thyroid gland, the trachea, or the larynx. On a CT scan, they often appears as solid, well-circumscribed masses with irregular borders, are heterogeneous, may contain calcifications, and are typically hypervascular on contrast-enhanced CTs, where the mass will enhance more than the surrounding tissue after the injection of contrast material [19]. Our patient presented a similar appearance to that described in the literature in the ultrasound and the CT evaluation of the cervical region. 

The management of parathyroid carcinoma before surgery includes reducing hypercalcemia through interventions, such as saline hydration, loop diuretics administration, calcitonin, bisphosphonates, and calcimimetic agents. This treatment aims to inhibit the bone release of calcium and increase the urinary excretion of calcium [20]. Surgery is the primary treatment for parathyroid carcinoma, with radiotherapy as a viable option for recurrent or metastatic disease [21,22]. 

It is essential to consider parathyroid carcinoma in the differential diagnosis of PTH-dependent hypercalcemia as complete resection of the tumor at the initial operation is associated with optimal outcomes. Diagnosis can be challenging, as it is rarely suspected preoperatively and may go undiagnosed during surgery. Postoperative management focuses on determining the success of the surgery and monitoring for symptomatic hypocalcemia. In cases of successful resection, PTH levels drop rapidly, often to undetectable levels. Local recurrence rates can be as high as 50%, and distant metastasis, particularly to the lungs, may occur and be signaled by recurrent severe hyperparathyroidism [1,2,3,4,5,6,7,8,9,10,11,12,13,14].

From a histopathological perspective, parathyroid carcinomas are large tumors with diameters exceeding 1.5 cm. They are often attached to adjacent tissues and can exhibit invasion of the contralateral thyroid lobe and cervical musculature. In 1973, Schantz and Castleman established criteria for microscopic diagnosis, including thick fibrous bands, elevated mitotic activity, and evidence of vascular and capsular invasion [23]. Differentiating between a carcinoma and an adenoma can be challenging, as both may exhibit similar features, such as nuclear pleomorphism, hyperchromatism, free tumor cells in blood vessels, mitosis, and giant cells. Diagnosing parathyroid carcinoma can be difficult due to its variable clinical presentations and examination findings. However, the histopathological examination of the present case revealed highly packed cells with a high degree of cellularity and round nuclei, displaying capsule infiltration and moderate mitotic activity, with no evidence of vascular invasion. Despite the capsule infiltration, the diagnosis was made early due to the absence of adjacent tissue invasion. Diagnosis of tissue can be aided by pathological examination using immunohistochemistry. This improves accuracy and can involve using proliferation markers, such as Ki-67 and cyclin D1, to differentiate between parathyroid carcinoma and adenoma. However, overlap among these tumor types limits the usefulness of this approach [24].

Regular follow-up is essential to monitor for the recurrence of parathyroid carcinoma, including monitoring of calcium and PTH levels every two months for the first five years and annually after that, along with annual cervical ultrasound scans. The 5-year and 10-year survival rates are reported to be around 77–100% and 66–80%, respectively, with a common local recurrence rate of 40–63% [25,26]. For suspicion of parathyroid carcinoma recurrence, modern imaging methods should be considered for periodic surveillance or when biochemical recurrence occurs with increased PTH or hypercalcemia [27].

In case of recurrent disease after a first operation, a 99mTc-sestamibi scan combined with single-photon emission computed tomography (SPECT) imaging can aid in accurate localization. Factors such as age, gender, time to first relapse, elevated serum calcium on recurrence, number of relapses, number of medications used to decrease calcium, difficulty in complete resection of the tumor, and aneuploidy of the tumor can also impact the prognosis of patients with parathyroid carcinoma. Lymph node involvement or distant metastasis to the lung, bones, and liver are associated with a higher recurrence rate, with up to 25% of patients potentially developing distant metastases during follow-up [28,29].

The preoperative therapeutic options, in this case, were adequate saline hydration, a loop diuretic, cholecalciferol, and bisphosphonates. Parathyroidectomy with serum calcium and PTH dosages was performed postoperatively. A close follow-up after two months that included serum calcium and PTH dosages and a cervical MRI scan was consistent with a relapse. Instead of a surgical reintervention, external radiation therapy was recommended, with good outcomes, with no abnormal parathyroid lesions, including in ectopic locations. 

In the last decade, it was demonstrated that [18F]FCh uptake of parathyroid adenomas strongly correlates with preoperative PTH serum level. The preoperative PTH level can predict the success of [18F]FCh-PET imaging in hyperparathyroidism, with higher lesion-to-background ratios expected in patients with high PTH, and also estimate the volume of parathyroid adenomas [30]. Another interesting study was the one performed by Liberini et al., which provided convincing evidence to support a potential correlation between 18F-choline uptake in parathyroid adenomas and their histological growth pattern and volumetric characteristics [31].

Considering both the low incidence of PC as well as the few cases reported in the literature, extensive multicenter studies are necessary for a more comprehensive imaging diagnosis, including other features as proposed by Liberini et al. and Alharbi et al. [30,31].

## 4. Conclusions

This case highlights the aggressive nature of parathyroid carcinoma and its management challenges. Due to the rarity of this disease, clinicians need to have a high index of suspicion and consider parathyroid carcinoma in the differential diagnosis of primary hyperparathyroidism, especially in the presence of a palpable mass or atypical imaging findings associated with a high level of PTH and hypercalcemia over 14 mg/dL. Adjuvant radiotherapy is recommended in relapse cases to improve the patient’s outcome. The management of thyroid carcinoma often requires a multidisciplinary approach, including the involvement of endocrinologists, surgeons, radiation oncologists, and medical oncologists. Close follow-up care is crucial to monitor for recurrent disease and manage any complications that may arise.

## 5. Teaching Points

Parathyroid carcinoma is an aggressive form of cancer, highlighting the importance of early detection and appropriate management strategies.Clinicians should maintain a high index of suspicion for parathyroid carcinoma when evaluating patients with primary hyperparathyroidism, especially if there are palpable masses or atypical imaging findings alongside elevated parathyroid hormone (PTH) and hypercalcemia levels.Adjuvant radiotherapy is recommended in relapse cases to improve patient outcomes and minimize the risk of disease progression.After parathyroidectomy, follow-up of patients with parathyroid cancer with 99mTc-sestaMIBI and 18-FDG PET/CT is necessary, especially in more aggressive and rapidly evolving forms.The management of parathyroid carcinoma often necessitates a multidisciplinary approach involving endocrinologists, surgeons, radiation oncologists, and medical oncologists.Close and regular follow-up care is essential to monitor for disease recurrence and promptly address potential complications.

These teaching points emphasize the importance of considering parathyroid carcinoma in the differential diagnosis of primary hyperparathyroidism, the significance of adjuvant radiotherapy in relapse cases, the multidisciplinary nature of managing parathyroid carcinoma, and the need for vigilant follow-up care.

## Figures and Tables

**Figure 1 reports-06-00040-f001:**
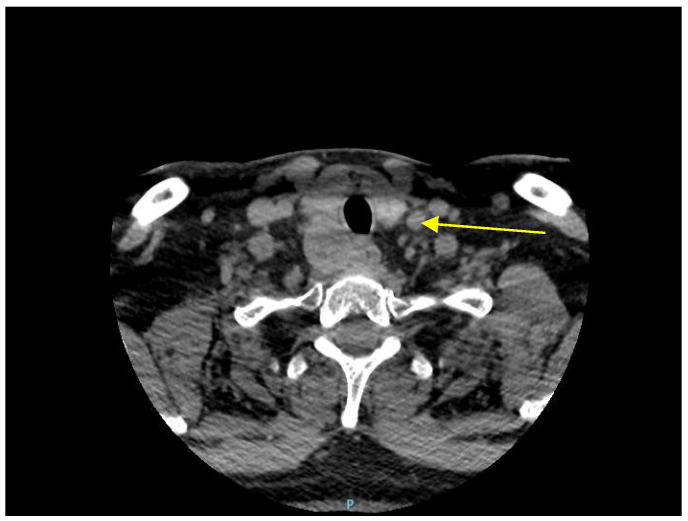
A cervical CT scan shows a 5 cm nodular mass in the right inferior parathyroid gland.

**Figure 2 reports-06-00040-f002:**
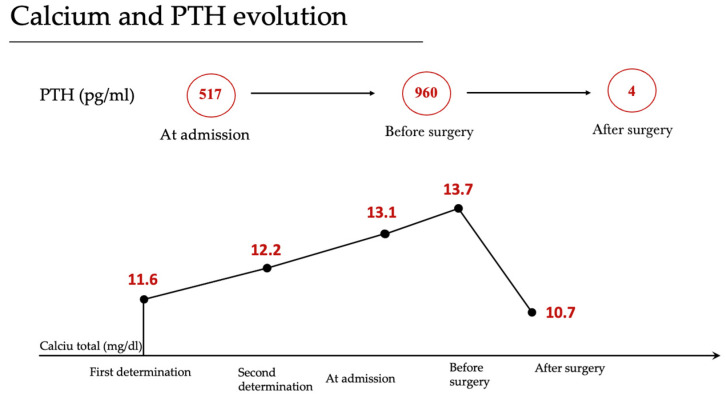
Calcium and PTH evolution.

**Figure 3 reports-06-00040-f003:**
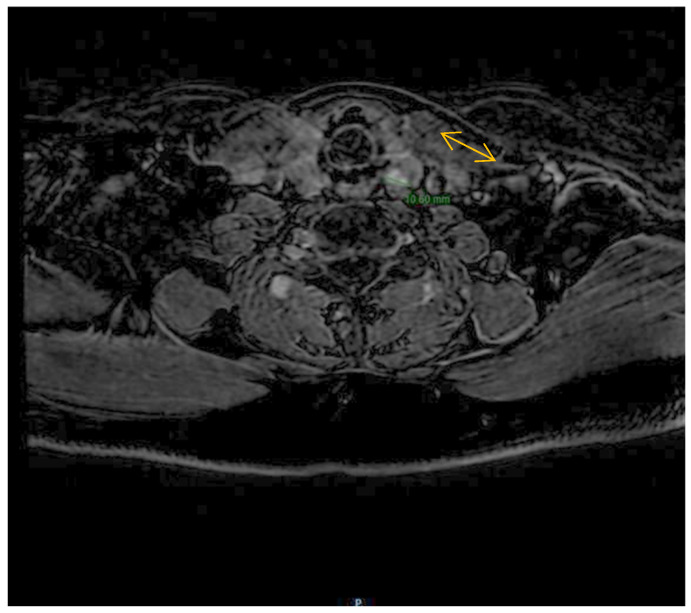
Neck MRI shows a new nodular mass of 11 mm adjacent to the left thyroid lobe at the base of the cricoid cartilage.

**Figure 4 reports-06-00040-f004:**
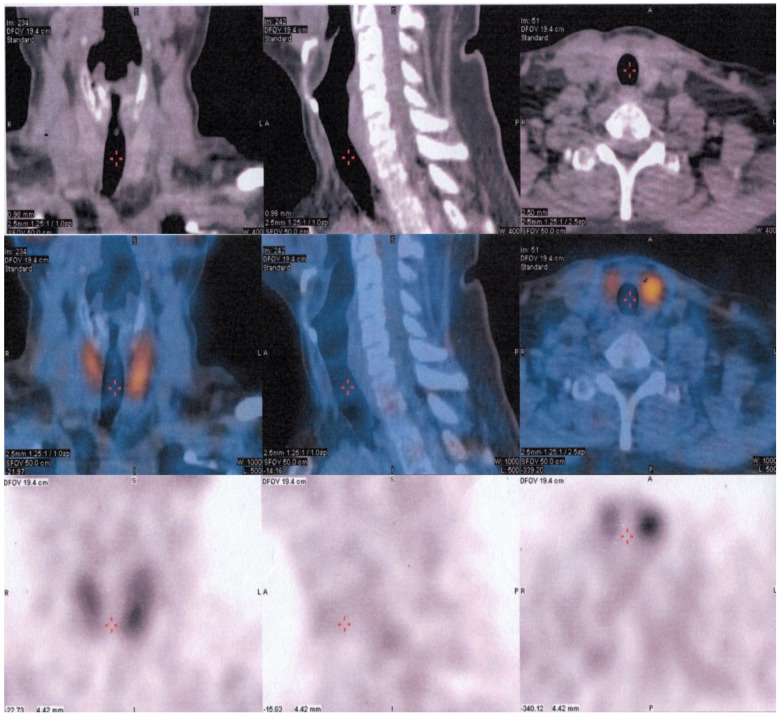
SPECT/CT combined with 99mTc-sestamibi scan without any abnormal parathyroid lesions, including in ectopic locations.

**Figure 5 reports-06-00040-f005:**
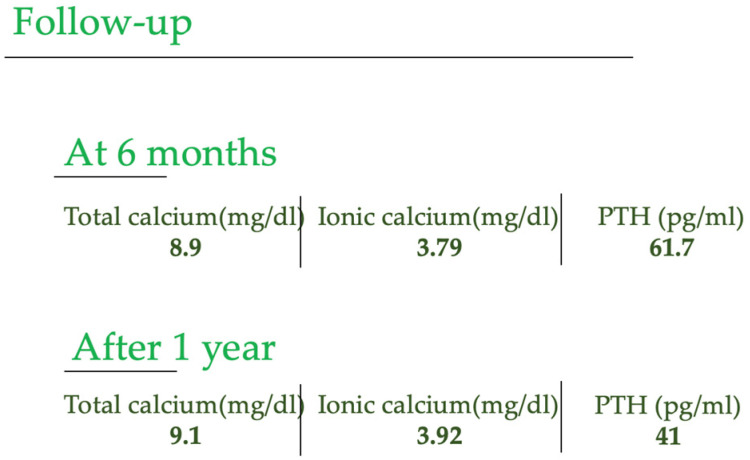
Calcium and PTH values during the follow-up period.

## Data Availability

More data available on request from the corresponding author.

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
