# Peer review of "Challenges in the Diagnosis of Parathyroid Cancer: Unraveling the Diagnostic Maze"

_reports, 2023, doi:10.3390/reports6030040_

Round 1

Reviewer 1 Report

The study evaluated a rare parathyroid cancer case and its recurrence. The results of the treatments and follow-up parameters are reported. Considering the study as benign in terms of diagnosis and treatment processes and the process followed afterward are explained in detail. I congratulate the authors for bringing this study to the literature. A history of parathyroid cancer is infrequent, and recurrence is much rarer. Surgical intervention and recurrence in a short period of 2 months is the prominent and exciting point of the study. 

What was the duration of your transition to the treatment, which starts with the patient's arrival at your clinic and after the correct diagnosis? Although his high serum calcium level indicates an increased risk of heart attack, how long did it take to decide on and perform the surgical procedure?

The rarity of the disease causes studies to be conducted even less frequently. The situation complicates the situation for the early diagnosis of parathyroid cancer directly. This study indicates the clinical process of parathyroid cancer and how far the issue still lags in early diagnosis. 

Author Response

Dear reviewer,

We appreciate your suggestions, and we hope the revised version of our manuscript will meet all the criteria for publication.

The study evaluated a rare parathyroid cancer case and its recurrence. The results of the treatments and follow-up parameters are reported. Considering the study as benign in terms of diagnosis and treatment processes and the process followed afterward are explained in detail. I congratulate the authors for bringing this study to the literature. A history of parathyroid cancer is infrequent, and recurrence is much rarer. Surgical intervention and recurrence in a short period of 2 months is the prominent and exciting point of the study. 

We appreciate your kind words.

What was the duration of your transition to the treatment, which starts with the patient's arrival at your clinic and after the correct diagnosis? Although his high serum calcium level indicates an increased risk of heart attack, how long did it take to decide on and perform the surgical procedure?

Within two weeks of the diagnosis, surgery was performed, depending on the surgeon's availability.

It took a week for all the necessary tests to be carried out to establish a correct diagnosis.  Treatment with diuretics (Furosemide), intravenous hydration, and alendronic acid were recommended to control the increase of calcemia. She was evaluated cardiologically, having no specific symptoms.

The rarity of the disease causes studies to be conducted even less frequently. The situation complicates the situation for the early diagnosis of parathyroid cancer directly. This study indicates the clinical process of parathyroid cancer and how far the issue still lags in early diagnosis. 

Reviewer 2 Report

Language correction is a must.

Language correction is a must.

Author Response

Language correction is a must.

Our manuscript has been revised.

Reviewer 3 Report

The authors reported a rare case of parathyroid cancer and presented the management of this case and insightful discussion about this area of scientific interest.

However, I offer several points for improvement: 

1. Please describe in more detail specific symptoms (e.g., which symptoms, how long, the reason for appointment, if regular or in emergency), if there were any other family or personal medical history. Please also specify in a nutshell, other relevant but normal laboratory tests (e.g., renal).

2.  Please explain how fast PTH and calcium levels increased from hospitalization till surgery (e.g., days?).

3. Please provide better imaging proof and for CT and MRI indicate the tumor with an arrow for more visibility.

4. Please explain radiotherapy decision better (Parathyroid tumors are generally not radiosensitive and recurrent tumor should also be resected; Local control may be improved with postoperative radiotherapy, but in this case report there was not adjuvant RT, but recurrent disease treatment). 

5. If the authors mention HRPT2/CDC73 gene they should explain the function of this gene and how mutation influence cancer disease (e.g., this is a tumor suppressor gene...).

6. Please provide more information about the relapse and the finality with this case report (e.g., how long did you follow-up this patient, what happened next, the recurrence was clinical, how laboratory tests were influenced?).

Author Response

Dear reviewer,

We appreciate your suggestions, and we hope the revised version of our manuscript will meet all the criteria for publication.

The authors reported a rare case of parathyroid cancer and presented the management of this case and insightful discussion about this area of scientific interest.

However, I offer several points for improvement: 

1.Please describe in more detail specific symptoms (e.g., which symptoms, how long, the reason for appointment, if regular or in emergency), if there were any other family or personal medical history. Please also specify in a nutshell, other relevant but normal laboratory tests (e.g., renal).

We revised the 2.1. section. The patient had no specific symptomatology and there was not a personal or familial history of endocrine disease. All the other tests performed were within the normal range ( hepatic, renal function).

“2.1.Patient Presentation

We present the case of a 51-year-old male patient referred to our clinic with no specific symptoms related to hypercalcemia or a family or personal history of endocrine disease. Physical examination revealed a palpable nodule in the right thyroid lobe region, quite delimited and firm, of approximately 3 cm, high blood pressure (150/100 mmHg), and a body mass index of 33.5 kg/m2. The patient was addressed to our clinic due to identifying an elevated calcemia level (as serum) of 12.78 mg/dL during routine blood investigations. The tests also showed elevated values at the next 2 repeat tests within 1 month.“

  1. Please explain how fast PTH and calcium levels increased from hospitalization till surgery (e.g., days?).

We have added a graphic.

  1. Please provide better imaging proof and for CT and MRI indicate the tumor with an arrow for more visibility.

We have revised our images.

  1. Please explain radiotherapy decision better (Parathyroid tumors are generally not radiosensitive and recurrent tumor should also be resected; Local control may be improved with postoperative radiotherapy, but in this case report there was not adjuvant RT, but recurrent disease treatment). 

The oncologist proposed Radiotherapy due to the post-op MRI appearance, which raised concerns. By performing CT spect, combined with Tm scintigraphy, together with the oncologist we demonstrated that radiotherapy was not necessary

  1. If the authors mention HRPT2/CDC73 gene they should explain the function of this gene and how mutation influence cancer disease (e.g., this is a tumor suppressor gene...).

We did not do genetic testing for our patient, due to the rapid evolution of the disease.

If the editor finds it relevant, we could add the following paragraphs regarding the genetic part. In our opinion, we do not find it adequate to add this to the introduction part, as there are already many excellent manuscripts published focusing on this matter.

“The HRPT2/CDC73 gene, also known as the hyperparathyroidism type 2 (HRPT2) or cell division cycle 73 (CDC73) gene, plays a critical role in the regulation of cell growth and proliferation. This gene is classified as a tumor suppressor gene, and its normal function involves encoding a protein called parafibromin. Parafibromin is part of the polymerase-associated factor (PAF) complex, which plays a vital role in transcriptional regulation and chromatin remodeling.

Mutations in the HRPT2/CDC73 gene can lead to the development of certain hereditary syndromes, such as hyperparathyroidism-jaw tumor syndrome (HPT-JT) and parathyroid carcinoma. In these syndromes, loss-of-function mutations in HRPT2/CDC73 result in the reduction or absence of functional parafibromin protein. This disruption impairs the PAF complex's proper functioning, leading to uncontrolled cell growth, impaired DNA repair, and increased susceptibility to the development of tumors, particularly in the parathyroid glands.

The connection between HRPT2/CDC73 gene mutations and cancer underscores the critical role of this gene in preventing tumorigenesis. Its tumor suppressor function is essential for maintaining cellular homeostasis and preventing the aberrant growth seen in cancer. As a result, a deeper understanding of the HRPT2/CDC73 gene's role and the mechanisms through which its mutations influence cancer disease is crucial for advancing our knowledge of cancer development and potentially developing targeted therapies for associated syndromes and malignancies.”

  1. Please provide more information about the relapse and the finality with this case report (e.g., how long did you follow-up this patient, what happened next, the recurrence was clinical, how laboratory tests were influenced?).

We have added a new section 2.7. Follow-up, in which we have detailed.

Reviewer 4 Report

The authors presented an interesting and well-written case concerning a patient with parathyroid carcinoma.

Minor suggestions:

- Page 6, line 222: "SPECT/CT combined with 99mTc-sestamibi scanning are extended"; this is the first appearance of the abbreviation "SPECT," so I suggest writing it in the extended form (and use only the abbreviation on page 7 line 273).

- Same goes for "FDG" and "PET" in the teaching point section.

- As a last suggestion: recent literature observed how high PTH levels might increase [18F]Fluorocholine uptake in parathyroid adenomas on PET/CT imaging (DOIs: 10.1007/s11307-018-1179-x ; 10.1097/RLU.0000000000003987; 10.1007/s12020-022-03280-9). Since parathyroid carcinomas are associated with very high PTH levels, I suggest the authors warrant prospective studies concerning the employment of choline PET/CT in the staging of parathyroid carcinomas in the discussion section.

Minor editing of English language required

Author Response

Dear reviewer,

We appreciate your suggestions, and we hope the revised version of our manuscript will meet all the criteria for publication.

The authors presented an interesting and well-written case concerning a patient with parathyroid carcinoma.

Minor suggestions:

- Page 6, line 222: "SPECT/CT combined with 99mTc-sestamibi scanning are extended"; this is the first appearance of the abbreviation "SPECT," so I suggest writing it in the extended form (and use only the abbreviation on page 7 line 273).

We revised it.

- Same goes for "FDG" and "PET" in the teaching point section.

Among the compounds labeled with 18-F for PET-CT, the most widespread use is radiolabeled glucose - 18F-FDG (fluorodeoxyglucose). This radiopharmaceutical has established itself in oncological pathology because most neoplastic cells are high consumers of glucose. A small proportion of the investigations using radiolabeled glucose are represented by other non-oncological indications, such as myocardial viability estimation, cerebral metabolism assessment, and evaluation of a limited number of inflammatory processes (eg, sarcoidosis). In oncological pathology, in most cases, the PET-CT examination is performed with 18F-FDG. The dose of radiopharmaceutical administered is calculated according to body weight, and the scan is performed 60-75 min post-injection.

 In the teaching point, we mentioned, " After parathyroidectomy, follow-up of patients with parathyroid cancer with 99mTc-sestaMIBI and 18-FDG PET/CT is necessary, especially in the more aggressive and rapidly evolving forms”.

We want to emphasize the importance of performing either 99mTc-sestaMIBI or 18-FDG PET/CT or both in the follow-up of patients operated for PC, proving through our case that MRI is not as reliable for monitoring these cases.

- As a last suggestion: recent literature observed how high PTH levels might increase [18F]Fluorocholine uptake in parathyroid adenomas on PET/CT imaging (DOIs: 10.1007/s11307-018-1179-x ; 10.1097/RLU.0000000000003987; 10.1007/s12020-022-03280-9). Since parathyroid carcinomas are associated with very high PTH levels, I suggest the authors warrant prospective studies concerning the employment of choline PET/CT in the staging of parathyroid carcinomas in the discussion section.

Our reference list has been updated following your recommendation.